# The Effect of *Brachionus calyciflorus* (Rotifera) on Larviculture and Fatty Acid Composition of Pikeperch (*Sander lucioperca* (L.)) Cultured under Pseudo-Green Water Conditions

Adrian A. Bischoff [1,*,†], Melanie Kubitz [1,†], Claudia M. Wranik [1], Laura Ballesteros-Redondo [1], Patrick Fink [2,‡] and Harry W. Palm [1]

1   Department of Aquaculture and Sea-Ranching, Faculty of Agricultural and Environmental Sciences, University of Rostock, Justus-von-Liebig-Weg 6, 18059 Rostock, Germany; melanie.kubitz@uni-rostock.de (M.K.); claudia.wranik@uni-rostock.de (C.M.W.); laura.redondo@uni-rostock.de (L.B.-R.); harry.palm@uni-rostock.de (H.W.P.)
2   Institute for Zoology, University of Cologne, Zuelpicher Strasse 47b, 50674 Koeln, Germany; patrick.fink@ufz.de
*   Correspondence: adrian.bischoff-lang@uni-rostock.de; Tel.: +49-381-498-3738
†   These authors contributed equally to this work.
‡   Present address: Department River Ecology and Department Aquatic Ecosystem Analysis and Management, Helmholtz-Centre for Environmental Research—UFZ, Brueckstrasse 3a, 39114 Magdeburg, Germany.

**Abstract:** A new cultivation system with the chlorophyte *Monoraphidium contortum* combined with a self-sustaining culture of the freshwater rotifer *Brachionus calyciflorus* was applied for *Sander lucioperca* (L.) larviculture. Survival, morphometrics, as well as fatty acid composition of pikeperch larvae were analyzed after a ten-day feeding period. By using the pseudo-green water technique with improved aeration and water movement at the surface, survival rates reached up to 94%, with a total larval length of $8.1 \pm 0.3$ mm and a specific length growth rate of up to 4.1% day$^{-1}$ for *S. lucioperca*. The biochemical composition of *B. calyciflorus* and especially its contents in C18 PUFAs and suitable n-3/n-6 ratios met the nutritional requirements of pikeperch larvae. The high abundance of highly unsaturated fatty acids (HUFAs) in the diet appeared to be less important in the first feeding due to a possible retention of essential fatty acids, which originate from the yolk sac reserves, at adequate levels. Exponential growth of microalgae and zooplankton under the applied conditions was most effective when stocking *M. contortum* five days and *B. calyciflorus* three days before adding the fish larvae. Appropriate timing and sufficient live feed density allowed a successful integration of *B. calyciflorus* into pikeperch larviculture. We hypothesize that feeding pikeperch larvae with a self-sustaining *Brachionus*-culture under pseudo-green water conditions with minor disruptions during larviculture will improve survival and growth. This system is a first step towards pikeperch larviculture inside recirculated aquaculture systems (RAS) under continuous feed supply with live feed within the same aquaculture unit.

**Keywords:** pikeperch *Sander lucioperca*; pseudo-green water technique; *Brachionus calyciflorus*; survival; growth; fatty acid composition

## 1. Introduction

The cultivation of pikeperch (*Sander lucioperca* (L.)), a high-priced commercial fish species, is gaining more and more attention [1–6]. In order to reach ecological and economical sustainability, an improvement of larviculture conditions is required, as the production of fry is still the major bottleneck in pikeperch aquaculture [6–10]. In order to improve pikeperch culture in terms of improved survival rates and hence better resource utilization, alternative live feed as well as improved culture techniques for intensive larval rearing seem necessary. To reduce water consumption and enhance control of rearing parameters,

recirculating aquaculture systems (RAS) were already applied for intensive larval rearing in Europe [11].

Lund and Steenfeldt (2011) [10] recorded higher survival rates, but a high output of pikeperch larvae relies on the use of cost-intensive *Artemia* nauplii and the inevitable need for *Artemia* enrichment with highly unsaturated fatty acids (HUFAs). As for the routine use of *Artemia* nauplii in marine aquaculture [12], this is also common in larval rearing of pikeperch [8]. However, the specific nutritional needs of pikeperch larvae are little understood [10], and it is assumed that *Artemia* nauplii cannot entirely cover the nutritional needs of freshwater species [11]. Periodical supplementation of live feeds and the short survival of *Artemia* nauplii in freshwater further reduce water quality [10].

As an alternative, the freshwater rotifer *Brachionus calyciflorus* was established as another live feed candidate for larval development of freshwater fish [13]. Especially in the rearing of fish with small and sensitive larvae, rotifers might increase larval survival and performance [13,14]. Awaiss et al. (1996) [15] reported improved survival, growth, and food conversion rates in larvae of perch (*Perca fluviatilis*) and gudgeon (*Gobio gobio*) fed with *B. calyciflorus* compared to dry or mixed feeds. However, unlike the mass cultivation of the marine rotifer *Brachionus plicatilis*, the commercial scale production of freshwater rotifers for larviculture so far is less established [11,16,17], and freshwater rotifers were only occasionally used in freshwater larval rearing [14]. This is despite the fact that freshwater rotifers do not lose their motility compared to saltwater rotifers in freshwater [6,18]. This sustained motility acts as a visual stimulus for the larvae and thus encourages them to forage. For the use of *Brachionus plicatilis*, it was further argued that a transfer of diseases and parasites should be avoided through the change of environment from saltwater to freshwater. However, based on Hennersdorf et al. (2016) [19], we argue that an early and low exposure to parasites might activate and improve the immune system of the fish larvae that consequently can react more quickly to future stressors. Furthermore, the plankton cultures used as feed are usually isolated cultures that have been cultivated over several life cycles without exchange with the natural environment. As a result, the risk of containing disease-causing pathogens or parasites and transferring them to the fish larvae is significantly reduced [20].

The positive effect of phytoplankton on larval rearing was previously established for Atlantic cod (*Gadus morhua*), turbot (*Scophthalmus maximus*), Atlantic halibut (*Hippoglossus hippoglossus*), European sea bass (*Dicentrarcus labrax*), striped mullet (*Mugil cephalus*), and summer flounder (*Paralichthys dentatus*) [21–26]. Beneficial effects may be due to the reduction of metabolites from fish and zooplankton [27]. Stimulating effects on the feeding behavior of European sea bass larvae (*Dicentrarchus labrax*) as well as digestive functions were described by Cahu et al. (1998) [22]. Additionally, the nutritional value of rotifers can be maintained through "green water" after addition of microalgae to the fish larval rearing tanks [24,28]. Reitan et al. (1993) [24] already reported increased biomass production, constant total lipid contents, as well as enhanced reproduction activity of rotifers in rearing systems of marine fish larvae supplied with microalgae. Papandroulakis et al. (2001) [29] developed a "pseudo-green" water technique based on a periodic supplementation of phyto- and zooplankton, combining the advantages of "clear water" and "green water". This method was already successfully applied in sea bream larviculture [29].

Due to the high abundance in the natural nursery grounds of pikeperch in Mecklenburg-Western Pomerania [30,31] and high reproduction and growth rates under laboratory conditions, the freshwater rotifer *B. calyciflorus* Pallas, 1766 (Gilbert, 1967) was applied as a live feed organism to establish the self-sustained zooplankton culture with high live feed density.

The aim of the present study was the establishment of a newly designed fish larviculture system to increase pikeperch larvae survival under aquaculture conditions. We hypothesize that feeding pikeperch larvae with a self-sustaining *Brachionus*-culture under pseudo-green water conditions with minor disruptions during larviculture will improve

survival and growth. The benefits of *B. calyciflorus* for pikeperch larviculture and an optimized run of the applied system are discussed.

## 2. Materials and Methods

### 2.1. Recirculating Aquaculture System for Pikeperch Larviculture

Three culture systems were assembled and used as replicates. Two independent, timely, separated experiments were carried out at the laboratory for Aquaculture and Sea-ranching, University of Rostock, from 17–27 May 2014 (experiment I) and 28 October–6 November 2014 (experiment II) (Table 1). The reason why two separate experiments were performed is that the reproducibility of the operating principle should be tested. Each culture system (total capacity of 90 L) consisted of two separated compartments, a larvae culture tank (filled with 30 L water), and a reservoir with 40 L, resulting, with the additional volume in the hoses and pipes, in a total volume of approximately 75 L per RAS during both experiments. In each system, the water circulated from the reservoir via a submersible pump, which did not damage *B. calyciflorus* during pumping, to the larvae culture tank (Figure 1). The water returned via an overflow from the larval culture tank, which was covered with a 200 μm mesh to hold back the fish larvae but allowed *Monoraphidium contortum* and *B. calyciflorus* to pass and circulate in the complete experimental system. Thus, a uniform *Monoraphidium contortum* and *B. calyciflorus* concentration could be assumed in all parts of the experimental system. The reservoir allowed *B. calyciflorus* reproduction and growth, nutrient uptake through microalgae, and water aeration.

**Table 1.** Experimental design for the two independent trials.

| | Experiment | |
| --- | --- | --- |
| | **I** | **II** |
| Duration | 7–27 May 2014 | 28 October–6 November 2014 |
| Experimental start | dph 0 | dph 0 |
| Experimental end | dph 10 | dph 10 |
| No. of replicates | 3 | 3 |
| System identification | I–III | I–III |
| Components | Culture tank, reservoir, pump | Culture tank, reservoir, pump |
| Stocking of *M. contortum* | 10 days prior to start * | 5 days prior to start |
| Stocking of *B. calyciflorus* | 3 days prior to start | 3 days prior to start |
| Stocking of *S. lucioperca* larvae | This was the start | This was the start |

* System I: 5 days prior to start.

### 2.2. Microalgae and Zooplankton Culture

The chlorophycean *Monoraphidium contortum* (Thuret) Komàrková-Legnerová (1969) (strain 47.80, obtained from SAG Culture Collection of Algae Göttingen, Göttingen, Germany) was grown in batch culture, inoculated in Erlenmeyer flasks of increasing volumes and finally cultured in 80 L carboys. For algal growth, 3 psu F/2-medium [32] was continuously added daily. The chosen light cycle was 16L:8D. For mass culture in the facilities of the University of Rostock, *M. contortum* was cultured with the same medium at room temperature (~20 °C) and the same light conditions and constantly aerated.

For the first experiment, each culture system was initially filled with a 75 L mixture of algae suspension (approximately $6.0 \times 10^5$ cells mL$^{-1}$), F/2 medium, and oxygen-saturated tap water ten days before starting the fish larvae experiment and continuously supplied with a set amount of F/2-medium ("fed-batch-process"). The algal cell density in the experimental systems was maintained at the initial level during the experiments by adding a continuous supply from an external phytoplankton-chemostat system, which was connected to the individual culture units as additional phytoplankton supply. The

maintenance of this cell density was indirectly controlled by absorption measurements (Hach-Lange, DR 3900, Düsseldorf, Germany) at a wavelength of 665 nm.

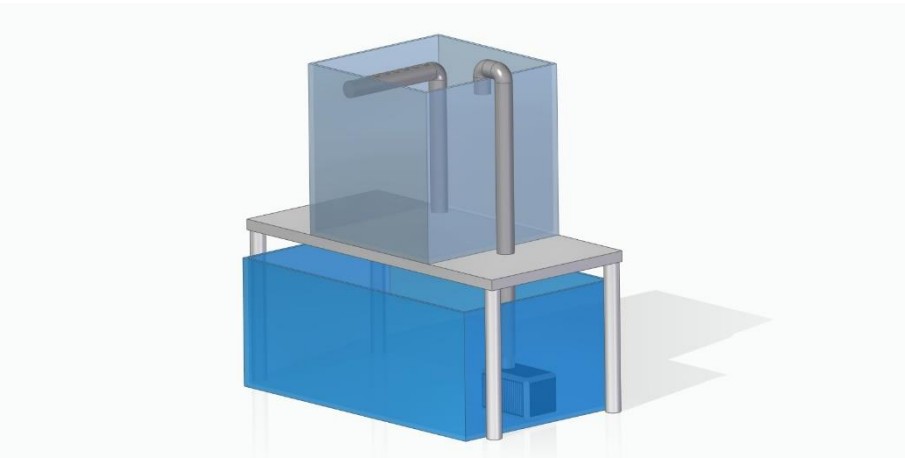

**Figure 1.** Scheme of the recirculating aquaculture systems with a 30 L larvae culture tank and a 40 L reservoir. The water circulated via an overflow from the larval rearing tank and returned via a submersible pump.

*B. calyciflorus* was obtained from Plankton-Zoo (www.plankton-zoo.de, Munich, Germany; accessed on 15 June 2013) and cultured as batch system in a 120 L cylindrical-conical polyethylene tank (PE) with daily addition of 1 to 2 L suspension of the chlorophyte *M. contortum* (including 3 psu F/2 medium). Culture was maintained at a salinity of 2.5 psu. No aeration or additional light was applied. Mixed size stages were fed ad libitum with *M. contortum* before inserting them into the experimental tanks.

Mixed stages of *B. calyciflorus* were stocked to each culture unit's reservoir three days before adding the fish larvae. *Brachionus* density was maintained at 5 Ind mL$^{-1}$ in the whole experimental system and an amount of 500 individuals as feed for each stocked fish larva per day was adjusted inside the tank. Therefore, each larvae culture tank was continuously supplied from the reservoir by the water circulation and from an additional external *Brachionus* culture tank to meet the nutritional requirements set above. This was necessary due to the insufficient reproduction rate of *B. calyciflorus* within the reservoir caused by the limited volume and the culture conditions that deviate from the optimal conditions of a monoculture. The externally added daily volume of the *Brachionus* culture depended on the density within the larvae culture tank. The daily addition was approximately 200 mL per system, which was thus the daily make-up of water. For quantification of the microlagae and *B. calyciflorus* growth in each culture system, five replicate samples of 5 mL were sampled daily and filtered through a 63 μm mesh. The filtrate was either transferred to a cuvette for photometric measurements or fixed with Lugol's solution for counting algae cells in a Fuchs–Rosenthal counting chamber (Marienfeld, Lauda-Königshofen, Germany) under a stereo light microscope (BX 53 Olympus). The extinction of the microalgae was determined with Hach–Lange photometer (DR 3900) at 665 nm. The retained *B. calyciflorus* on the mesh was fixed with Lugol's solution and was counted using a stereo light microscope (SZX10 Olympus, Hamburg, Germany).

### 2.3. Pikeperch Larval Culture

Pikeperch larvae were obtained from a commercial fisherman in Hohen Sprenz (Germany, Mecklenburg-Western Pomerania, 53°54′47″ N; 12°11′49″ E) and from the research facility Hohen Wangelin of the Institute for Fisheries of the State Research Centre Mecklenburg-West Pomerania. Fish used as broodstock for pikeperch experiments, originating from wild catches in Lake Hohen Sprenz and Lake Müritz, had a length range of 60

to 75 cm and were assumed to spawn repeatedly. The breeders were reared in net cages in Lake Hohen Sprenz and were fed a natural diet (site-specific lake fish). Since the fisherman manages Lake Hohen Sprenz, he also regularly stocks commercially important species. This also includes pikeperch. When the spawning temperature of approximately 16 °C was reached, the breeders were transferred to tanks, and these tanks were equipped with spawning substrate to which the females could attach the eggs. After spawning, the eggs were removed from the tanks, deglutinated, and transferred to incubators for hatching. Fish used as broodstock for pikeperch experiments from Hohen Wangelin were offspring of the system and were also repeated spawners.

The larvae hatched at a water temperature of 16 °C, and directly thereafter (dph = 0), they were transferred to the laboratory facility of the University of Rostock via car in a specialized fish transport box. We applied a temperature increase during several hours to acclimate the larvae to laboratory conditions. Fish larvae were randomly distributed into our three setups for both individual experiments.

Pikeperch larvae were stocked at lower densities compared to Szkudlarek and Zakęś (2007) [2] to ensure a high live feed concentration per fish larvae and to minimize interspecific competition. The initial larval stocking density was 10 Ind $L^{-1}$ inside the units, which resulted in a total larval number of 300 individuals per culture tank.

The average culture conditions (±SD) were $19.3 \pm 0.9$ °C temperature, a salinity of $0.4 \pm 0.2$ psu, oxygen saturation of $100.9 \pm 7.1\%$, pH of $6.9 \pm 0.2$, and a redox potential of $159.4 \pm 14.8$ mV. The average dissolved nutrients were $0.12 \pm 0.15$ mg $L^{-1}$ $NH_4^+$, $0.071 \pm 0.085$ mg $L^{-1}$ $NO_2^-$, $4.6 \pm 3.1$ mg $L^{-1}$ $NO_3^-$, and $0.77 \pm 0.42$ mg $L^{-1}$ $PO_4^{3-}$. Larval rearing was performed under daylight with a photo period of 11:13 h (L:D). In the systems, pikeperch larvae were reared with the pseudo-green water technique [29] by adding daily phytoplankton and zooplankton to the culture tanks (see above). The survival rate of the pikeperch larvae was determined at the end of each experiment (dph 10) by individual counting of all remaining larvae.

Fish larvae were sampled for morphological analyses from the individual tanks and stored in 70% ethanol until measurement approximately 15 to 20 days after sampling. At the start of the experiment, 45 larvae were initially sampled from the transport boxes. At the end of the feeding period of experiment I, 38 larvae from system I, 22 larvae from system II, and 14 larvae from system III were sampled, according to the survival rates of the replicates. For experiment II, the final samples for the morphological analyses consisted of 10 individual larvae per unit. The total length of the fish larvae was measured by using a stereo light microscope (SZX10 Olympus, Hamburg, Germany) connected to a UC30 digital camera (Olympus, Hamburg, Germany) and the software package cellSens Dimension 1.6 (Olympus Soft Imaging Solutions, Hamburg, Germany). Therefore, fish larvae were individually placed under the stereo light microscope, and the longest distance between the tip of the head and the tip of the tail was recorded.

The specific growth rate (SGR) [% $d^{-1}$] of the pikeperch larvae was calculated, with the total length of the larvae according to Jørgensen (1990) [33], applying the formula

$$SGR = (\ln(L_t/L_o) \times t^{-1}) \times 100 \tag{1}$$

where $L_t$ and $L_o$ represent the total length of the larvae at time t and time t = 0.

### 2.4. Sampling of Fish Larvae for Fatty Acid Analyses

In order to investigate the supply of pikeperch larvae with highly unsaturated fatty acids (HUFAs) during exclusive feeding with *B. calyciflorus* under pseudo-green water conditions, fatty acid analyses of pikeperch larvae were conducted. Five fish larvae samples were taken at the beginning of the experiment (dph 0) and 4 samples at the end (dph 10). One sample encompassed between 10 and 25 individuals per tank, depending on the individual body mass of the larvae, to collect sufficient material for the fatty acid extraction. The larvae were collected and killed in accordance with the applicable laws and regulations, and the larvae for fatty acid analyses were shock frozen immediately after death.

The detection of larval dry mass and fatty acid analyses were carried out for larvae of replicate I of the first experiment because microalgae and zooplankton did not develop as expected in the other replicates, and therefore, high mortality rates of the fish larvae were observed during experiment I. Due to limited capacities concerning the fatty acid analyses and freezer storage problems, no samples were taken from the second experiment.

Samples were taken, and fatty acid analyses were conducted according to the method by Windisch and Fink (2018) [34]. Lipid extraction was conducted by immersing homogenized tissue in a dichloromethane: methanol mixture (2:1/*v:v*) for at least 12 h. Thereafter, fish samples were sonicated and centrifuged for 5 min ($4500 \times g$). After taking up the lipid phase quantitatively, the solvent was evaporated to dryness under a stream of nitrogen gas (5.0 purity grade) at 40 °C. In the following, fatty acids were trans-esterified with 5 mL of 3 N methanolic HCl at 70 °C for 20 min to their fatty acid methyl ester (FAME) derivatives [35]. FAMEs were extracted with $2 \times 2$ mL iso-hexane and after evaporation of the samples under a stream of nitrogen gas were finally dissolved in 100 μL isohexane. For gas chromatographic analyses, 1 μL of the sample solution were measured on a 6890 N GC System (Agilent Technologies. Waldbronn, Germany) equipped with a DB-225 capillary column (30 m length, 0.25 mm inner diameter, 0.25 μm film thickness). Temperature programming with helium (5.0 purity grade) as carrier was applied (flow rate of 1.5 mL min$^{-1}$). Samples were injected using a programmable temperature vaporizer injector (solvent vent mode) using the following temperature program: injector and FID temperatures 200 °C, initial oven temperature 60 °C for 1 min, followed by a 20 °C min$^{-1}$ temperature ramp to 150 °C, then 7 °C min$^{-1}$ to 220 °C, followed by a final 14 min at 220 °C. FAMEs were detected by flame ionization and identified by comparing the retention times of detected peaks with those of reference compounds and quantified using two internal standards (tricosanic acid methyl ester, 23:0 ME and nonadecanic acid, 19:0 ME) and previously established calibration functions for each individual FAME.

### 2.5. Statistical Analyses

Statistical analyses were performed by using IBM SPSS Statistics, Version 22. For the test of normal distribution, the Shapiro–Wilk test was applied. To test the homogeneity of variance the Levene's test was used. To analyze differences between means an analysis of variance (ANOVA) was performed. In case significant results were obtained, post hoc tests followed either Tukey–Kramer (with variance homogeneity) or Dunnett T3 post hoc tests (without variance homogeneity). To analyze differences between two groups, a *t*-test was applied. As non-parametric tests, either a Mann–Whitney U-test or a Kruskal–Wallis analysis of variance (ANOVA) was chosen. All significance levels α were set to 0.05.

## 3. Results

### 3.1. Growth of Monoraphidium Contortum and Brachionus Calyciflorus in the Culture Units

Each culture unit of the first experiment was inoculated with *M. contortum* 10 days before starting the fish larval experiment at dph 0. Due to a yellowish discoloration of the culture water in system I, a complete water exchange was conducted five days before stocking the fish larvae. At the beginning of the fish larvae experiment, the highest cell density of $2.5 \pm 0.5 \times 10^6$ cells mL$^{-1}$ of *M. contortum* could be detected in replicate III, while in replicate systems I and II, lower algal cell densities of $2.2 \pm 0.1 \times 10^5$ cells mL$^{-1}$ and $3.2 \pm 0.3 \times 10^5$ cells mL$^{-1}$ were observed (Figure 2). After stocking pikeperch larvae, the green water of replicates II and III also turned yellow-brown one day post hatch, but a complete water exchange was no longer possible to avoid additional stress on the pikeperch larvae and to maintain comparability. For experiment II, the three individual culture units were inoculated with microalgae five days prior to stocking of the fish larvae at dph 0.

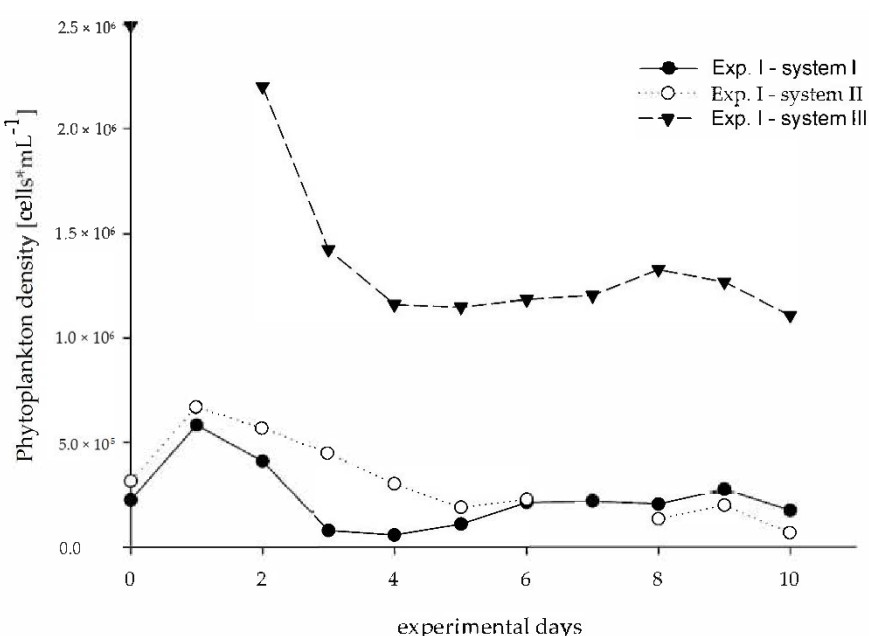

**Figure 2.** Density of *Monoraphidium contortum*, displayed as daily means ($\pm$SD in cells mL$^{-1}$) for each culture system during the pikeperch larval rearing of experiment I (day post hatch 0–10) (*n* = 5 per sampling).

Mixed stages of *B. calyciflorus* were stocked three days before starting experiment I. After three days, at the beginning of the fish larval experiment I, system I reached a live feed density of 2.8 $\pm$ 0.3 Ind mL$^{-1}$, while a density of 0.2 $\pm$ 0.1 Ind mL$^{-1}$ and 0.0 Ind mL$^{-1}$ of *B. calyciflorus* was detected in samples taken from systems III and II, respectively (Figure 3). Despite external addition of *B. calyciflorus*, an increase in its concentrations was only detected in replicate I, while the mean densities in systems II and III were close to zero (0.1 $\pm$ 0.2 Ind. mL$^{-1}$ and 0.1 $\pm$ 0.1 Ind. mL$^{-1}$, respectively). In system I, *M. contortum* as well as *B. calyciflorus* grew exponentially, with a peak of *M. contortum* at day one post hatch (5.8 $\pm$ 0.2 $\times$ 10$^5$ cells mL$^{-1}$) and the highest *B. calyciflorus* density at day four post hatch (12.7 $\pm$ 0.5 Ind mL$^{-1}$).

During experiment II, all cultures of *M. contortum* developed as expected and according to the conversion with the absorption data exceeded cell densities above 1.2 $\pm$ 0.5 $\times$ 10$^6$ cells mL$^{-1}$, which, on the basis of previous experience from other plankton experiments, has been assumed to be sufficient to support a good zooplankton growth. The initial *B. calyciflorus* densities of the three individual systems were 0.03, 0.08, and 0.06 Ind. mL$^{-1}$ (Figure 4). Due to these low zooplankton concentrations, additional feeding was required at experimental day 2. *B. calyciflorus* concentration peaked in the systems at 7.9 Ind. mL$^{-1}$, 33.4 Ind. mL$^{-1}$, and 4.2 Ind. mL$^{-1}$ for the three individual culture units (Figure 2) at experimental days 8, 9, and 8, respectively. Comparing the concentration of *B. calyciflorus* within the three separate systems, it is noticeable that there is a clear variability between system II and the other two systems. This manifests itself in concentrations of over 3o Ind. mL$^{-1}$ in system II compared to concentrations of about 6–8 Ind. mL$^{-1}$.

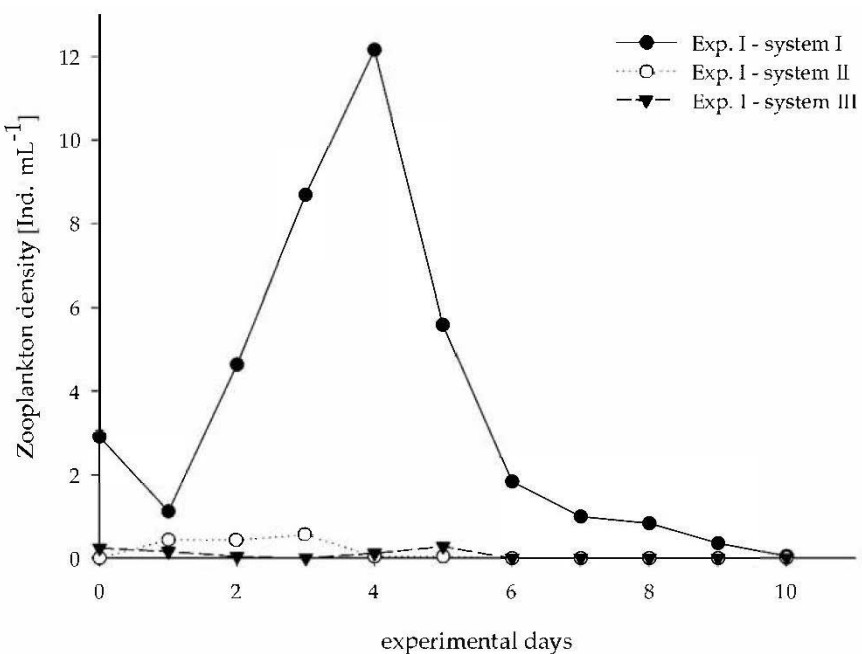

**Figure 3.** Density of *Brachionus calyciflorus*, displayed as daily means (±SD in Ind. mL$^{-1}$) for each culture system during pikeperch larval rearing of experiment I (day post hatch 0–10) (*n* = 5 per sampling).

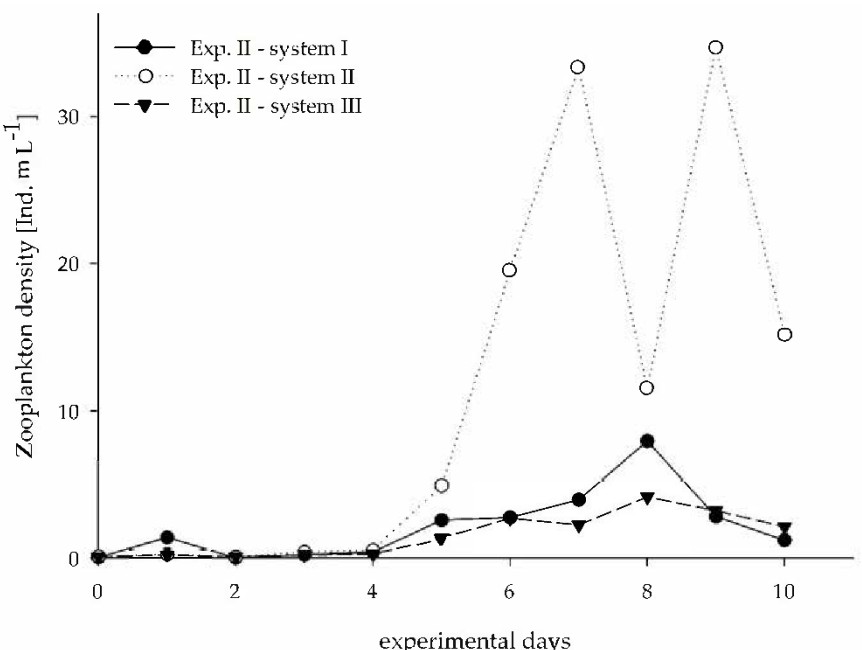

**Figure 4.** Density of *Brachionus calyciflorus*, displayed as daily means (±SD in Ind. mL$^{-1}$) for each culture system during pikeperch larval rearing of experiment II (day post hatch 0–10) (*n* = 5 per sampling).

### 3.2. Survival, Growth, and Total Fatty Acid Content of Pikeperch Larvae

After a ten-day feeding period with *B. calyciflorus* under pseudo-green water conditions in experiment I, the highest survival rate of 94% was observed in system I, whereas the survival rates in systems II and III were considerably lower (Table 2). The lowest larval survival of 5% was observed in system III, followed by a survival rate of 7% in system II. For experiment II, survival rates ranged from 25% (system I) to 64% (system II) and 74% (system III) (Table 2).

**Table 2.** Morphometric characteristics and survival of pikeperch larvae at dph 0 and dph 10 (Experiment I: initial sample: $n$ = 45; ten days post hatch: system I, $n$ = 38; system II, $n$ = 22; system III, $n$ = 14. Experiment II: initial and final $n$ = 10). Superscript capitals indicate statistically different groups in terms of size after hatching and at the end of the experiment. Superscript lower-case letters indicate statistical differences between the three treatments at the end of the experiment.

| | Total Length (mm) | Specific Growth Rate—SGR (% d$^{-1}$) | Survival Rate (%) |
|---|---|---|---|
| Experiment I: | | | |
| After hatch | 5.4 ± 0.1 [A] | | |
| Culture system I | 8.1 ± 0.3 [Ba] | 4.1 | 94 |
| Culture system II | 6.6 ± 0.6 [Bb] | 2.0 | 7 |
| Culture system III | 5.9 ± 0.1 [Bb] | 0.9 | 5 |
| Experiment II: | | | |
| After hatch | 5.1 ± 0.2 [C] | | |
| Culture system I | 6.0 ± 0.2 [Dc] | 1.7 | 25 |
| Culture system II | 6.8 ± 0.3 [Dd] | 2.9 | 64 |
| Culture system III | 6.4 ± 0.2 [Dcd] | 2.3 | 74 |

In each replicate of the first experiment, a significant (Kruskal–Wallis, $p < 0.001$ and $p = 0.019$) increase of the total length (TL) of the pikeperch larvae was detected during the first ten days, and the TL of pikeperch larvae in system I was significantly (Kruskal–Wallis, $p < 0.001$ and $p = 0.013$) higher compared to larvae of the other two culture units (Table 2). The larval dry mass increased significantly (ANOVA, $p < 0.001$), while the sum of total fatty acids (TFAs) as well as the amount of fatty acids per unit dry weight decreased significantly (ANOVA, $p < 0.001$) within the first ten days (Table 3). Due to low survival rates, fatty acid profiles as well as dry masses (DM) could not be determined for larvae of systems II and III.

**Table 3.** Total fatty acids (µg mg$^{-1}$ DM) and individual larval dry mass (µg) of pikeperch larvae at dph 0 and dph 10 from system I (initial sample: $n$ = 5; ten days post hatch: replicate I, $n$ = 4). Superscript lower-case letters indicate statistical differences between larval hatching and the end of the experiment. The letters are to be read for the columns of the table, respectively.

| | Total Length (mm) | Dry Mass (µg) | Total Fatty Acids (µg mg$^{-1}$ DM) |
|---|---|---|---|
| Days post hatch: 0 | 5.4 ± 0.1 [a] | 113.8 ± 7.0 [c] | 110.4 ± 2.2 [e] |
| Days post hatch: 10 | 8.1 ± 0.3 [b] | 270.1 ± 3.5 [d] | 39.0 ± 0.9 [f] |

*3.3. Fatty Acid Profile of Pikeperch Larvae at Hatching and at 10 dph*

Within the PUFAs, DHA was the most abundant fatty acid in yolk-sac larvae, followed by the two n-6 PUFAs, linoleic acid, and ARA. The content of the n-3 PUFAs α-linolenic acid and EPA was lower at hatching, but due to a high DHA level, the sum of n-3 PUFAs was higher than n-6 PUFAs in newly hatched pikeperch larvae (Table 4). According to the measured concentrations, in descending order, docosahexaenoic (22:6 n-3; DHA), oleic (18:1 n-9), palmitoleic (16:1 n-7), palmitic (16:0), linoleic (18:2 n-6), arachidonic (20:4 n-6; ARA), eicosapentaenoic (20:5 n-3; EPA), α-linolenic (18:3 n-3), and vaccenic (18:1 n-7) acid were the dominant (i.e., concentrations $\geq 5$ µg mg$^{-1}$ DM) fatty acids in newly hatched larvae (Table 5).

**Table 4.** Initial and final fatty acid ratios obtained from the pikeperch larvae from system I of the experiment I.

| Fatty Acid Ratios | dph 0 | dph 10 |
|---|---|---|
| n-3/n-6 | 2.0:1.0 | 2.4:1.0 |
| DHA/EPA | 3.7:1.0 | 5.1:1.0 |
| ARA/EPA | 1.1:1.0 | 1.5:1.0 |
| 18:3n-3/18:2n-6 | 1.0:1.7 | 1.0:1.1 |

**Table 5.** Dominant fatty acids (means $\pm$ SD in $\mu$g mg$^{-1}$ DM) of pikeperch larvae at days post hatch 0 ($\geq 5$ $\mu$g mg$^{-1}$ DM) and at days post hatch 10 ($\geq 1$ $\mu$g mg$^{-1}$ DM) as well as percentages of the corresponding initial content (initial sample: $n = 5$; after ten days post hatch: system I, $n = 4$). A "$-$" indicates a significant decrease in the fatty acid concentration, whereas a "+" indicates a significant increase in the fatty acid concentration. Data are organized in descending quantities.

| | Larvae at Days Post Hatch 0 | | | Larvae at Days Post Hatch 10 | | |
|---|---|---|---|---|---|---|
| | Fatty Acids | $\mu$g mg$^{-1}$ DM | | Fatty Acids | $\mu$g mg$^{-1}$ DM | % of Initial Content |
| 1. | 22:6n-3 | 20.5 $\pm$ 1.7 | 1. | 16:0 | 7.9 $\pm$ 0.6 *$^-$ | 81 |
| 2. | 18:1n-9 | 18.6 $\pm$ 0.6 | 2. | 22:6n-3 | 5.9 $\pm$ 0.1 *$^-$ | 29 |
| 3. | 16:1n-7 | 12.1 $\pm$ 0.4 | 3. | 18:1n-9 | 4.9 $\pm$ 0.3 *$^-$ | 26 |
| 4. | 16:0 | 9.8 $\pm$ 0.5 | 4. | 18:0 | 4.9 $\pm$ 0.1 *$^+$ | 121 |
| 5. | 18:2n-6 | 8.9 $\pm$ 0.1 | 5. | 18:2n-6 | 3.2 $\pm$ 0.1 *$^-$ | 36 |
| 6. | 20:4n-6 | 6.0 $\pm$ 0.3 | 6. | 18:3n-3 | 3.0 $\pm$ 0.1 *$^-$ | 58 |
| 7. | 20:5n-3 | 5.6 $\pm$ 0.4 | 7. | 20:4n-6 | 1.7 $\pm$ 0.0 *$^-$ | 28 |
| 8. | 18:3n-3 | 5.2 $\pm$ 0.1 | 8. | 20:5n-3 | 1.2 $\pm$ 0.0 *$^-$ | 21 |
| 9. | 18:1n-7 | 5.2 $\pm$ 0.2 | 9. | | | |

* Significantly different contents compared to the initial value.

Compared to newly hatched larvae, there was an increasing n-3/n-6 ratio, DHA/EPA ratio, ARA/EPA ratio, and $\alpha$-linolenic acid:linoleic acid ratio after ten days of feeding on *B. calyciflorus* (experiment I, system I). Each ratio was significantly (*t*-test, $p < 0.001$ and $p = 0.001$) different compared to the ratios observed for newly hatched larvae.

Due to a general decrease of fatty acid contents (per dry matter) during the run of the experiment, the term dominant fatty acid was adjusted for fatty acids recorded with concentrations $\geq 1$ $\mu$g mg$^{-1}$ DM. Consequently, ten days post hatch, palmitic acid (16:0) was the dominant ($\geq 1$ $\mu$g mg$^{-1}$ DM) fatty acid, followed by DHA, oleic acid (18:1 n-9), stearic acid (18:0), $\alpha$-linoleic acid, linolenic acid, ARA, and EPA. For the dominant fatty acids, in relation to dry mass, the level of each PUFA as well as of palmitic acid decreased significantly (ANOVA, $p < 0.001$, and Kruskal–Wallis $p = 0.003$) within the first ten days, while a significant increase (ANOVA, $p < 0.001$) of stearic acid per dry mass was detected. DHA remained the most abundant and linoleic acid the second-most abundant PUFA ten days post hatch. In terms of C18 PUFAs, the $\alpha$-linolenic acid/linoleic acid ratio was nearly balanced ten days post hatch contrary to newly hatched larvae. Further, the proportion of DHA as well as of ARA increased in relation to EPA compared to newly hatched larvae.

Among MUFAs, each of the dominant MUFAs in newly hatched larvae was utilized significantly (ANOVA $p < 0.001$ and Kruskal–Wallis, $p = 0.007$). While the final content of oleic acid was still high ten days post hatch, the level of palmitoleic acid and vaccenic acid was lower than 1 $\mu$g mg$^{-1}$ DM, and only 4.1% and 18.2%, respectively, of the initial contents were detected at the end of the experimental period.

## 4. Discussion

The overall idea of the experimental setup was separation of the fish larval rearing and the cultivation unit of *B. calyciflorus* into different compartments of one single RAS under constantly circulating water. The reservoir should enable undisturbed *B. calyciflorus*

reproduction, thereby achieving a self-sustained live feed culture, which should serve also as a feed reservoir for the fish larvae. Additionally, control of the culture conditions inside the culture system, without intervention into the larval rearing tank, was possible in the reservoir. The green water with *Monoraphidium contortum* ensured water turbidity and enhanced the quality of the culture water through nutrient extraction by microalgae, and the primary production served as food for the freshwater rotifer *Brachionus calyciflorus*.

It was demonstrated that the newly developed culture system for larviculture of pikeperch achieved survival rates that were above average by using *B. calyciflorus* as first live feed (between 5 and 94% at 10 dph). The maximum survival rate of 94% exceeds most of the published rates for pikeperch and perch larvae [2,14,36–38]. The only known exception is the study of Lund and Steenfeldt (2011) [10], where at a temperature of 18 °C and a larval stocking density of 28 Ind L$^{-1}$, no mortality was observed until days post hatch 16. A final survival rate of 91% was detected 21 days post hatch. However, in our experiments, the survival rates between the triplicates varied, and 3 of the 6 systems had much lower survival rates, in the range of 5–25%. This may be a consequence of the small system size that causes variation between replicates. Other influencing factors in addition to the small system volume could also have been the action by several employees or the different locations of the test systems although each system was individually equipped with lighting and a water and air pump. Nevertheless, it demonstrates that very high survival rates can be achieved under the conditions we describe.

The average total length and the SGR of pikeperch larvae after a culture period of ten days in system I of experiment I were 8.1 mm and 4.1% day$^{-1}$ and in system II of experiment II reached 6.8 mm and 2.9% day$^{-1}$ (Table 2) and was comparable to larval lengths observed in Szkudlarek and Zakęś (2007) [2] 11 days post hatch and even higher than total lengths observed in pikeperch larvae 12 days post hatch fed different enriched *Artemia* [39] or 11 days post hatch fed with *Brachionus plicatilis* at different salinities [18]. Imentai et al. (2019) [18] found that at a salinity of 2 psu, the total length was about 5.75 mm and the SGR 10.0 ± 5.4% day$^{-1}$, which is not consistent with our data, which show, for example, a length of 8.1 ± 0.3 mm at dph 10 and a specific growth rate of 4.1% day$^{-1}$. Consequently, our data demonstrate an advantage of using the freshwater rotifer, *B. calyciflorus*, in comparison to saltwater live feed, such as *B. plicatilis* and *Artemia* spec. Applying *B. plicatilis* in freshwater, it is noticeable that they die after a relatively short time and are no longer available as feed. Consequently, there is less feed available. Compensating this effect of lower feed quantity with increased feed portions, leads to a deterioration in water quality, as dead *B. plicatilis* sink to the bottom and are quickly decomposed. This aspect, which does not occur with *B. calyciflorus*, was interpreted by us as an advantage of *B. calyciflorus* over *B. plicatilis*.

*Artemia* eggs that have not been sieved and are therefore not offered as micro-artemia cannot be ingested by the pikeperch larvae in the first days of external feeding due to their size, as the mouth opening of the pikeperch larvae is simply smaller than he *Artemia* nauplii. This fact was also interpreted as an advantage of *B. calyciflorus*, which can be ingested be the pikeperch larvae. The use of micro-artemia mitigates the size argument, but *Artemia* spec. does not fully meet the nutritional requirements of pikeperch larvae. This was also considered an advantage for *B. calyciflorus* due to the fact that B. calyciflorus can be ingested and digested by the pikeperch larvae.

However, our observed growth clearly depended on the timing of the zooplankton peak that could be achieved in the utilized setup. The observed length of 5.1–6.8 mm during experiment II was lower compared to the best-achieved values in experiment I (8.1 mm), which reached the *B. calyciflorus* peak about 4–5 days post hatch. In experiment II, the peak was reached not before days 8 or 9 post hatch, which was too late for an optimal feed supply and resulted in a reduced growth. It should be mentioned here that the biology of *B. calyciflorus* played an important role here. During both experiments, no account was taken of egg-bearing females when determining the concentration of *B. calyciflorus*, so the sudden increase in system II during experiment II can be attributed to the hatching of existing eggs. The rapid decrease can then be attributed to the feeding of the pikeperch larvae.

Unfortunately, due to the late Brachionus peak, this increased feed availability could not be used for increased growth rates. Yet, such variations should not be over-interpreted as long as the concentrations do not fall below a critical threshold.

*B. calyciflorus* is known to multiply its population within few days. Rico-Martínez and Dodson (1992) [40] demonstrated that it increased from 25–104 Ind. mL$^{-1}$ in 2 days at 30 °C with $5 \times 10^6$ cells mL$^{-1}$ of *Chlorella vulgaris*. Consequently, our freshwater rotifer with specimen numbers of 2.8 and 12.7 Ind. mL$^{-1}$ after 4 days in experiment I–system I had similar population growth rates with less feed concentration (*M. contortum* $5.8 \times 10^5$ cells mL$^{-1}$), demonstrating that *B. calyciflorus* grew well during our experiment I. Moreover, the peak in *B. calyciflorus* abundance matched with the start of exogenous feeding, which is known to be around 5 dph for pikeperch larvae [6]. On the other hand, a lack of enough adequate microalgae for *B. calyciflorus* occurred during the first experiment in systems II and III, resulting in an inadequate live feed density for pikeperch larvae. During the second experiment, maximum *B. calyciflorus* densities appeared at the middle or the end of the experimental period. This delay reduced survival and growth rate in comparison to experiment I–system I. According to Bischoff et al. (2018) [5], the delay in zooplankton production should not exceed 160 to 170 day-degrees, when all internal reserves must be compensated to avoid increased mortality rates.

*M. contortum* in experiment I–system I reached a density of $5.8 \pm 0.2 \times 10^5$ cells mL$^{-1}$ at 1 dph. The microalgae at the beginning had no nutrient limitation and was at the exponential phase when *B. calyciflorus* where stocked. Preliminary tests and other experiments also clearly demonstrated that *M. contortum* can be considered a suitable feeding alga for the establishment of an artificial feed chain for pikeperch larvae. Further manuscripts to prove this statement are in preparation. Own studies showed that the availability of nutrients in relation to the N:P ratio can enhance the production of PUFAs by the microalgae, which was supported by literature [41,42]. Although this is species-specific, the phosphate limitations that occur at the stationary phase reduce the PUFAs production, being the best N:P ratio for *Monoraphidium* 0.153:1 to 1.53:1 [43–45]. Our microalgae culture in system I was at the appropriate phase and nutrient profile in the first experiment, when we stocked the *B. calyciflorus*. As a consequence, *B. calyciflorus* grew exponentially and thus allowed for a high *B. calyciflorus* density at the start of the exogenous feeding of pikeperch larvae. Therefore, our study demonstrates that stocking of *Monoraphidium contortum* in such small-scale systems should be done, considering the size of the culture system and the initial stocking density. In our case, this means it should not commence earlier than 3 days before the stocking of *B. calyciflorus* and five days before stocking the fish larvae. These results were confirmed by the second experiment, where all three individual microalgae cultures were similar. The results indicate that the synchronization of *M. contortum* and *B. calyciflorus* growth is essential to increase survival and growth rates of pikeperch larvae by using life feed.

Individual dry mass of pikeperch larvae during the first experiment (system I) increased significantly after ten days post hatch. The dry mass was comparable to dry weights of pikeperch larvae after eight days post hatch in the study of Lund and Steenfeldt (2011) [10] and lower compared to dry weights observed 12 days post hatch in the study of Lund et al. (2012) [39]. Differences in growth might be explained by different culture conditions, different times of measurement, and the use of *Artemia* in the study of Lund et al. (2012) [39] from day 4 post hatch onwards with a higher biomass gain compared to rotifers. Total fatty acids (TFAs) per individual larvae as well as per mg of dry mass decreased significantly, which was also observed in several studies of other fish species [7,46–49]. Bischoff et al. (2018) [5] reported a decrease in fatty acids to about 20% of the original concentration, which was even lower compared to the present study, which reached a fraction of 35%. This can be explained by the yolk sac consumption during the first days before the start of the exogenous feeding. In perch, after feeding with *B. calyciflorus*, the TFAs was 24.2 % [15]. Therefore, we assume our results of about 35% of the original fraction or 39 μg mg$^{-1}$ DM as an adequate content in TFAs.

In relation to the PUFAs, our data of n-3/n-6 ratio in pikeperch larvae before and after feeding *B. calyciflorus* are similar to n-3/n-6 ratios that could be observed in pikeperch eggs and larvae from wild, mature breeders [10]. Therefore, our results of n-3/n-6 (2:1) were assumed to reflect the natural ratio for pikeperch larvae and may indicate that *B. calyciflorus* meets the requirements of pikeperch in relation to these PUFAs. DHA was the second most abundant fatty acid and the most abundant PUFA in larvae ten days post hatch, and 29% of the initial content could be recovered in larvae. Similar observations could be made for ARA, for which 28% of the initial content could be detected in larvae after ten days feeding *B. calyciflorus,* and the final content was higher compared to the final EPA level which was 21% of the initial content. As a result, the DHA/EPA ratio was also considerably different in pikeperch at 10 dph, but the EPA/ARA ratio remained similar. The fatty acid profile of the *B. calyciflorus* culture at different conditions [15,50,51] could explain the high levels of DHA that were found in Awaiis et al. (1996) [15] for perch as well as in our experiment. Furthermore, other studies observed that in starved pikeperch larvae, a minimum of 30% of the initial DHA content that was retained in larvae after ten days without exogenous feeding [5,46]. Therefore, DHA and consequently the HUFA ratios may be less affected by the diet and could be regulated through selective retention of specific HUFAs from larval reserves. Hence, we assume from our results that (dietary) enrichments of n-3 HUFAs are not crucial in the early development of pikeperch larvae, contrary to other studies in pikeperch FA requirements [52]. Nonetheless, such a statement must be backed up by scientific experiments and data. Therefore, further experiments have been and are being conducted, and the data already obtained are currently being prepared for further publications that will provide a more accurate picture of the importance of the dynamics of fatty acids in the early development of pikeperch larvae.

Both C18 PUFAs linoleic acid and α-linolenic acid were well present in pikeperch larvae after the ten-day feeding period with *B. calyciflorus*, and the final content of both C18 fatty acids was higher compared to, e.g., ARA and EPA. It is well-established that C18 PUFAs are highly abundant in freshwater ecosystems [53–55] and that there is a general demand in freshwater and diadromous fish for C18 PUFAs [56,57]. Due to significantly higher levels of C18 PUFAs in the eggs of wild compared to the eggs of cultivated pikeperch breeders in the study of Khemis et al. (2014) [58], a significant role of C18 PUFAs and a natural demand can be assumed in pikeperch. Despite this natural demand, observations during previous studies (unpublished data) provided evidence that the content in α-linolenic and linoleic acid as well as their ratio in fish larvae are highly associated with the diet of pikeperch breeders [58] and with the content of larval first feeds [10,14,15].

Different studies have tried to understand the interaction of microalgae, *B. calyciflorus*, and fish larvae in terms of fatty acids [15,50,51]. In Kennari et al. (2008) [51], *B. calyciflorus* had a α-linolenic/linoleic ratio of approximately 1:1 with less α-linolenic acid compared to the respective diet (*Chlorella*) and the linoleic acid content similar to the microalgae. This suggests that *B. calyciflorus* also needs and uses α-linolenic acid for growth. Furthermore, in our experiment, due to the water supplementation and thus also nutrient supply, microalgae were supposed to be not limited in nutrients, producing more n-3 and providing enough α-linolenic acid to rotifers, as was also observed by Jensen and Verschoor (2004) [50], where *B. calyciflorus* had a α-linolenic/linoleic acid ratio of 2.3:1 when feeding on microalgae without nutrient limitation. The α-linolenic acid was in a similar range as the applied microalgae diet, and compared to linoleic, it showed twice the amount. This means that, even if rotifers used α-linolenic acid to live, the supplied amount was definitely enough and further pointed out the importance of the microalgae diet for the FAs content of the fish [59]. In our system, rotifers should have filled the needs of pikeperch in terms of their α-linolenic/linoleic acid ratio of 1:1.1. This is in contrast to previous observations of Bischoff et al. (2018) [5] for starved larvae (1:3). This could also explain the higher levels of DHA since the pikeperch larvae could have converted α-linolenic acids into DHA, which was previously discussed [15] and is a known process in freshwater fishes [60]. Moreover, a diet rich in α-linolenic and poor in HUFAs, as *B. calyciflorus* [15,50,51], seem to enhance the

ß-oxidation [61]. However, one study showed no evidence of this process in pikeperch [52] and thus reinforces our hypothesis that pikeperch larvae retain HUFAs. Nevertheless, we demonstrate that *B. calyciflorus* fills the nutritional requirements of pikeperch larvae producing a balanced $\alpha$-linolenic acid/linoleic acid ratio (1:1) that might be optimal in pikeperch during early development.

## 5. Conclusions

Our study confirmed that the pseudo-green water technique in RAS is adequate for pikeperch larviculture, promoting high survival and growth rates by the use of an adequate timing/matching of *M. contortum* and *B. calyciflorus* with the start of exogenous feeding. Moreover, our results indicate that *B. calyciflorus* is an adequate live feed for the first 10 days post hatch of pikeperch larvae, and the interaction with the microalgae plays an important role. The microalgae culture conditions determine the fatty acid composition of *B. calyciflorus* as a diet for pikeperch, and thus, a good timing of microalgae and zooplankton results in a high live feed density with an adequate nutrient profile at the start of larval feeding. In the absence of algal nutrient limitation, the biochemical composition of *B. calyciflorus* and especially their content in C18 PUFAs and suitable n-3/n-6 and $\alpha$-linolenic acid/linoleic acid ratios seems to meet the nutritional requirements of pikeperch larvae. Furthermore, the dietary HUFA composition seems to be less important for pikeperch during the first ten days after hatching.

**Author Contributions:** Conceptualization, A.A.B., M.K., C.M.W. and H.W.P.; methodology, A.A.B., M.K., C.M.W. and P.F.; validation, A.A.B., M.K., P.F. and H.W.P.; formal analysis, A.A.B. and M.K.; investigation, A.A.B., M.K. and C.M.W.; resources, H.W.P.; data curation, M.K. and L.B.-R.; writing—original draft preparation, M.K. and A.A.B.; writing—review and editing, A.A.B., M.K., C.M.W., L.B.-R., P.F. and H.W.P.; visualization, A.A.B. and M.K.; supervision, P.F. and H.W.P.; project administration, A.A.B. and C.M.W.; funding acquisition, H.W.P. All authors have read and agreed to the published version of the manuscript.

**Funding:** This research was part of the project "Entwicklung eines Zooplankton-Reaktors zur Unterstützung der Fischlarvenaufzucht relevanter Zielfischarten in Mecklenburg-Vorpommern" and was partly funded through the European Fisheries Found (VI-560/7308-4).

**Institutional Review Board Statement:** The animal study protocol was approved by the State Office for Agriculture, Food Safety, and Fisheries Mecklenburg-Western Pomerania—Veterinary Services and Agriculture—(protocol code 7221.3-2-021/14; date of approval: 2 May 2014).

**Informed Consent Statement:** Not applicable.

**Data Availability Statement:** Not applicable.

**Acknowledgments:** We would like to thank Carsten Kühn and Gregor Schmidt from the Institute for Fisheries of the State Research Centre Mecklenburg-West Pomerania for the supply with pikeperch larvae. We also thank Katja Preuß from the Workgroup Aquatic Chemical Ecology at the University of Cologne for her assistance and support during the fatty acid analyses. Finally, we would like to thank Erwin Berchtold for his help with the visualization.

**Conflicts of Interest:** The authors declare no conflict of interest.

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
