# Peer review of "The Effect of Brachionus calyciflorus (Rotifera) on Larviculture and Fatty Acid Composition of Pikeperch (Sander lucioperca (L.)) Cultured under Pseudo-Green Water Conditions"

_sustainability, doi:10.3390/su14116607_

Round 1

Reviewer 1 Report

This paper investigated the effect of Brachionus calyciflorus (Rotifera) on larviculture and fatty acid composition of pikeperch (Sander lucioperca (L.)) cultured under pseudo-green water conditions. Overall, the manuscript will contribute positively to the corresponding field. However, the description of the whole paper is somewhat lengthy and not clear enough in expression. Also, there are some major concerns that the authors should pay attention to:

I feel confused about the feeding time, whether ten-day feeding is sufficient to see the effect?

Introduction: How much information should I give up front? Ask yourself before starting writing. In addition, the introduction is too long, please make it concise. Line 47-55, these descriptions are too specific, please rewrite it.

The experimental design is also somewhat complicated. Please use figure or table for readers easily understood.

P value should be italic and capital.

Linen 91, please check the citation is right? [29][32].

Line 109: “Aquaculture System”, a capital letter needed?

Line 256: This sentence doesn't need two spaces in the first line, does it? please check the format of this paper.

Line 383: The description such as (means ± SD in μg*mg-1 DM), (initial sample: n = 5; after ten days post 385 hatch: system I, n = 4), should be put into the footnotes.

The format of references was not consistent. Citation style guidelines are often published in an official handbook containing explanations, examples, and instructions. Please read the instructions in the submission system and correct it.

Author Response

Thank you for the review of our manuscript. 
We have tried to incorporate your comments and advice as best we could and to answer the questions. 
The attached PDF gives the answers to all three reviewers. 

Reviewer 2 Report

Very interesting work well planned and well written.

Just 4 minor issues:

  • Better to use always Brachionus calyciformis instead of zooplankton throughout the text because this species was the only one considered.
  • Point 2.3. Try to explain a little better how pikeperch larvae were obtained: how eggs were incubated?; how the larvae were collected? I do not understand how a commercial fisherman could has given the larvae.
  • The results of the first column in Table 3 are repeated in Table 2.
  • Either in results and discussion, careful must be used when affirm “high survival” because this was only achieved in system I of experiment I (one replicate) with the better conditions of microalgae and prey density. Are there any other possibilities to explain replicate differences besides “small system size”?

Author Response

(The authors gave the same response as above.)

Reviewer 3 Report

plae find attached

Author Response

(The authors gave the same response as above.)

Round 2

Reviewer 1 Report

None.

Author Response

Thank you for your effort. We appreciate it and you helped to improve our manuscript.

Reviewer 3 Report

Please find attached

Author Response

Thank you again for your effort.

It looks like that the file submitted was renamed but it should contain the replies to the reviewer.

Round 3

Reviewer 3 Report

Comments to authors rebuttal 2. round:

Regarding larval survival: the authors continue to compare observed survival rates  in their work to a published study (Yanes-Roca, 2018).  However, the comparison flaws, as the study by Yanes-Roca measured survival 17 dph while in the present manuscript the trial ended at dph 10 !  -  I agree that mortality is usually higher from dph 1-10, than  dph 10-17,  but other factors like cannibalism is rising from app. dph 15 depending on growth and temp.

-  this is an example of not using a proper reference and thus statements, which are not completely right 

regarding the requirement of LC - PUFAs and the limited capacity to convert EPA to DHA, I will recommend authors to read the work by Reis et al: Comp. Biochem. and Physiol. Part B, 246-247 (2020)